# Circadian Rhythms in Tongue Features

**DOI:** 10.3390/jcm13123549

**Published:** 2024-06-17

**Authors:** Jihye Kim, Woosu Choi, Keun Ho Kim, Dong-Hyun Nam

**Affiliations:** 1Digital Health Research Division, Korea Institute of Oriental Medicine, Yuseongdae-ro 1672, Yuseong-gu, Daejeon 34054, Republic of Korea; wisdom00311@kiom.re.kr (J.K.); harrius@kiom.re.kr (W.C.); rkim70@kiom.re.kr (K.H.K.); 2Department of Biofunctional Medicine and Diagnosis, College of Korean Medicine, Sangji University, Sangjidae-gil 83, Wonju 26382, Republic of Korea

**Keywords:** tongue features, tongue color, tongue coating, circadian rhythm

## Abstract

(1) **Background:** The aim of this study was to investigate the circadian rhythms of tongue features according to the effects of physiological phases over a 24 h period. (2) **Methods:** Fifteen healthy participants aged 20 to 69 years were recruited. The participants did not have current chronic diseases or past diseases and had to meet the inclusion and exclusion criteria. The participants stayed at the Gil Hospital for a duration of 2 nights and 3 days. On the first day, at 18:00, they consumed their allocated portions of food and water and then completed a questionnaire. At approximately 21:00, their tongue images were acquired using a computerized tongue image acquisition system, following which they slept for 8 h, commencing at 23:00. Measurements were taken from 07:00 through 21:00 on the second day, and the final acquisition was taken at 07:00 on the following morning, resulting in a total of eight images. The circadian rhythm was authenticated and quantified utilizing the single cosinor analysis, a technique for periodic regression analysis for fitting a 24 h cosine curve. (3) **Results:** Cosinor analysis revealed that all tongue features were significantly related to circadian rhythm. (4) **Conclusions:** The results of this study may be important for considering the time of day at which the tongue is observed and tongue status is evaluated.

## 1. Introduction

Circadian rhythms encompass physical, mental, and behavioral changes synchronized with the human biological clock, which operates on roughly a 24.2 h cycle. These rhythms are primarily influenced by exposure to light and darkness in an organism’s environment and are pervasive across various life forms, including animals, plants, and even certain microscopic organisms [1].

The circadian rhythm is governed by the suprachiasmatic nucleus or nuclei of the hypothalamus, influenced by both exogenous and endogenous factors. Exogenous factors include physical activity, mental and emotional stress, eating habits, caffeine intake, vitamin D3 synthesis, and environmental factors such as light, temperature, and sound. Endogenous factors involve the sympathetic and parasympathetic nervous systems, plasma concentrations of melatonin and adrenaline, vascular resistance, vasoconstriction, blood volume, and the renin–angiotensin–aldosterone system. The importance of circadian rhythm has gained recognition in recent years due to its reflection of bodily states, including the prognosis of certain pathologies, thereby playing a significant role in medicine and healthcare [2].

Many studies have attempted to measure interactions between the circadian cycle and homeostatic processes and reported that physical parameters such as body weight, temperature, and blood pressure change according to circadian rhythms [3,4,5,6].

Oral factors, including the salivary flow rate, oral temperature [7], volatile sulfur compounds, salivary composition, and oral microbiome, also showed circadian variations. Previous investigations have shown that the salivary flow rate fluctuates with the circadian cycle. It has been suggested that the unstimulated flow rate may be at its maximum in the mid-afternoon [8,9]. Similarly, a previous study reported that the oral temperature reached a maximum in the early evening and a minimum in the early morning, with a maximum–minimum range of 0.9 °C [10], and oral temperature showed a circadian rhythm with a 24 h periodicity [8,9]. Schneyer et al. [11] and Lear et al. [12] reported that the salivary flow rate decreases markedly during sleep; on the other hand, the salivary flow rate increases markedly during intake. Shinjiro Koshimune, DDS. reported that this study showed an extreme reduction in the resting saliva flow rate when the amount of tongue coating increased [13]. Dawes, C. showed that unstimulated whole saliva showed significant circadian rhythms of high amplitude with an acrophase at 15:26. Dawes, C. studied the circadian rhythm of unstimulated and stimulated saliva. As a result, the unstimulated and stimulated saliva showed circadian rhythms that were significant for the group as a whole flow rate. Dawes, C. reported that salivary compositions in unstimulated saliva showed circadian rhythms in the concentrations of sodium, potassium, magnesium, chloride, and inorganic phosphate but not in protein or calcium, whereas salivary compositions in stimulated saliva showed circadian rhythms in sodium, potassium, calcium, magnesium, chloride, and inorganic phosphate, while protein does not exhibit such circadian rhythm [8,9]. Ikawa, K. et al. reported that the levels of volatile sulfur compounds exhibited circadian changes and drastically changed according to meals and brushing. The volatile sulfur content was highest early in the morning, significantly reduced after meals, and gradually increased until the next meal [14]. Miyazaki, H. et al. found no notable variances in volatile sulfur compounds between males and females across all age groups. Within each age cohort, oral malodor measurements peaked in the late morning group, followed by the late afternoon group, with the lowest levels observed in the early afternoon group. Additionally, a significant correlation was noted between the levels of volatile sulfur compounds and tongue coating [15]. Although this is not a study on circadian changes, there is research on variations in tongue features, such as tongue color and tongue coating thickness, according to the menstrual cycle [16,17]. Kim, J., et al. revealed that the CIE Lab color value in the tongue coating area and the tongue coating thickness of primary dysmenorrhea patients during the menstrual phase were significantly lower than those of the non-dysmenorrhea patients [16]. Hsieh, S.-F. et al. reported that tongue color changes during the menstrual cycle and suggested that there are differences in tongue color between the follicular and luteal phases among childbearing women [17].

In theory and previous studies, oral factors influenced by the biological clock, which can be measured or evaluated, can serve as indicators for assessing the phase, period, and amplitude of the circadian rhythm. Unfortunately, no reports have been published regarding changes in tongue features such as tongue color or tongue coating thickness according to circadian rhythm.

Based on the abovementioned changes in physiological parameters within a circadian cycle, we speculated that tongue features such as tongue body color, tongue coating color, and tongue coating thickness might change within a circadian cycle in healthy participants. The aim of this study was to elucidate the circadian variations in tongue features in healthy participants.

## 2. Materials and Methods

### 2.1. Study Design and Ethical Statement

This study was designed as a prospective and observational feasibility study and was performed at Gachon University Gil Medical Center, Incheon, Republic of Korea, from September 2015 to September 2016. All clinical data were obtained from healthy laboratory participants after written informed consent was provided. The study was approved by the Institutional Review Board of the Gachon University Gil Medical Center, Republic of Korea (authorization number: GDIRB2015-229), and was registered with the CRIS (registration number: KCT0001679).

### 2.2. Participants and Enrollment

Fifteen healthy laboratory participants aged 20 to 69 years were recruited by advertising using a poster and a banner displayed in the hospital and in a local subway. The participants did not have current chronic diseases or past diseases, according to a doctor’s notes and laboratory tests.

To be eligible for the study, the enrolled healthy participants had to meet the following criteria: (1) aged from 20 to 69 years; (2) body mass index (BMI) ranging from 18.0 to 25 kg/m^2^; and (3) signed an informed consent form after a full explanation of the purpose of the study was given.

The exclusion criteria were as follows: (1) had an endocrine disease, cardiovascular disease, severe liver disease, blood disease, respiratory disease, digestive system disease, or brain disease; (2) had a mental disorder; (3) were pregnant or nursing; (4) had a history of drug or alcohol abuse; (5) had a sensitive reaction to general laboratory tests; (6) were current smokers; or (7) refused to provide informed consent. Participants who were taking medications or had diseases that influence oral physiology were excluded from this study.

Participants meeting any of the following criteria were excluded from the final statistical analysis: (1) experiencing adverse events (AEs) or serious AEs necessitating withdrawal from the study, and (2) having any measurement obtained inconsistently with the standard operating procedures of the investigational devices.

### 2.3. Study Procedures

The participants stayed at Gil Hospital for a duration of 2 nights and 3 days and visited the laboratory for clinical study. On the first day, at 18:00, they consumed their allocated portions of food and water before completing the inclusion and exclusion questionnaire. Around 21:00, their tongue images were obtained using a computerized tongue image acquisition system (abb. CTIS, TAS-4000, Korea Institute of Oriental Medicine, Daejeon, Republic of Korea), following which they slept for 8 h, commencing at 23:00. Measurements were conducted at 3 h from 07:00 through 21:00 on the subsequent day, and the final measurements were taken at 07:00 on the following morning, resulting in a total of eight tongue images per one participant. Table 1 shows the experimental procedures.

Foods were eaten at 08:00, 12:00, and 18:00. The dietary intake was normocaloric, with water being consumed during meals at a rate of 20 mL/kg/day, and the water naturally occurring in the food was not considered. Approximately 2 L of water was consumed over a 24 h period, excluding the water content of the food.

The consumption of highly colored foods and sweets can affect the tongue color. One should ensure that the patient has not consumed foods of this kind just prior to the examination. Spicy foods such as pickles, cayenne peppers, and curries tend to redden the tongue for a short time after consumption [18].

Each participant was evaluated at approximately the same time in a measurement room designed with temperature and humidity control systems, ensuring consistency across evaluations and minimizing the influence of biological rhythms. The room temperature was maintained at 25 ± 1 °C during the 24 h period.

### 2.4. Measurement

The investigational device in this clinical study was developed to meet ISO 20498-1—CTIS—Part 1: General requirement and ISO 20498-2—CTIS—Part 2: Lighting environment (Figure 1).

The CTIS consists of tongue positioning part, lighting part, image acquisition part, and data processing part (example: analysis software). To minimize color errors, the image acquisition part was calibrated using a color checker (X-Rite, Grand Rapids, MI, USA) before capturing the tongue images. The accuracy of color calibration was assessed by calculating measurement errors between gold standards (true color values) and measurement color values of color checker. A low measurement error indicates the high accuracy of the color measurement [19,20].

The image acquisition part can capture the tongue image at resolutions of 1080 × 1440 pixels and automatically adjust white balance, exposure time, and gain settings. The lighting part should meet the following conditions: an illuminance of 500 lux, a color temperature over 5000 Kelvin, a color rendering index greater than 90, and a min/max ratio of illuminance distribution under 0.9.

To obtain the most reliable measurement results, the participant received instructions prior to the first measurement [16]. Two hours prior to tongue image acquisition, the participant should not practice any oral hygiene, eat, or drink coffee. Additionally, participants were instructed to adhere to the following criteria: (1) refraining from using any oral rinse for 1 week preceding the experiment day; (2) avoiding consumption of caffeine, alcohol, milk, and smoking for 24 h prior to the experiment; and (3) abstaining from eating and drinking for a minimum of 8 h before the experiment, although they were permitted to drink water up to 3 h before the experiments. These measures were implemented to minimize the influence of various factors.

Furthermore, participants were instructed to adhere to the following guidelines: (1) refrain from using any oral rinse or breath freshener for one week preceding the experiment day; (2) avoid consuming caffeine, alcohol, milk, or smoking for 24 h prior to the experiment; and (3) abstain from eating and drinking for a minimum of 8 h prior to the experiment, with the exception of water intake permitted up to 3 h before the experiments. These measures were implemented to mitigate the impact of various factors on the study outcomes.

The participants were asked to extend their tongues for only 15–20 s. Prolonged extension of the tongue tends to redden it, with particularly quick reddening at its tip. If a longer examination is needed, one can ask the participant to withdraw the tongue, close the mouth, and then extend it again. This can be performed several times without affecting the tongue color.

### 2.5. Extraction and Calculation of Tongue Features

The outcomes of this study were tongue features, including tongue body color, tongue coating color, and tongue coating percentage, as measured by the CTIS.

A whole tongue area was automatically segmented using the combined polar edge method and the gradient vector flow snake technique [21]. For incorrect segmentations, both manual and automatic segmentation were performed repetitively. A dark region in the tongue root area because of low illumination intensity and high-luminance regions caused by light reflection from saliva were removed using thresholding with CIE L* color values, as these colors could introduce errors into the color histogram analysis. The region of the tongue was extracted from the captured image, distinguishing the tongue coating area from the tongue body area based on the variance between their respective RGB color values. Subsequently, obtained RGB color values of each pixel in both areas were converted to CIE Lab color values. The mean CIE Lab color values of the tongue body and the tongue coating area were then computed. The tongue coating percentage was determined by calculating the ratio of the pixel count of the tongue coating area to the total pixel count of the entire tongue area. The process for calculating these tongue indices is illustrated in Figure 2. The tongue image extraction methods and image analysis methods for tongue features in CTIS have been described in detail in previous studies [21,22].

The CIE L* color value in the tongue body area is suitable for representing paleness intensity. A lower CIE L* color value indicates a paler tongue compared to a redder tongue. On the other hand, calculated CIE L* color value in the tongue coating area is appropriate for representing the amount of tongue coating or tongue coating thickness. Meanwhile, the CIE a* color value in tongue body area is suitable for representing red intensity [22]. A lower CIE a* color value indicates a paler red tongue compared to a redder tongue. The tongue coating percentage served as the index for estimating the amount of tongue coating or tongue coating thickness. The tongue coating percentage was used as the index to estimate the amount of tongue coating or tongue coating thickness [23].

### 2.6. Statistical Analysis

The demographic characteristics were summarized by the mean and standard deviation for each continuous variable. Differences in the demographic characteristics between the male and female groups were compared using an independent two-sample *t* test. A normality test was performed for each variable before the independent two-sample *t* test using the Kolmogorov–Smirnov and Shapiro–Wilk tests. If the assumption of normality was not confirmed, the Mann–Whitney test was used.

The single cosinor technique was initially pioneered and extensively employed by Franz Halberg for analyzing biological rhythms, particularly in cases involving short time series and sparse data when prior information is available. Cosinor analysis entails a regression approach that fits one or more cosine curves to the data, either separately or concurrently, with the aim of minimizing the sum of squares of the differences between the actual measurements and the fitted model (known as residuals) over a specified period. From this model, one obtains, for the period considered, an estimate of (i) the rhythm-adjusted mean or midline estimating statistic of rhythm (M, MESOR), defined as the average value of the curve fitted to the data; (ii) amplitude (A), defined as half the height of oscillation in a cycle approximated by the fitted cosine curve (the difference between the maximum and the MESOR); (iii) acrophase (ø, a measure of phase), the lag from a defined reference time point (e.g., local midnight or another significant reference) to the crest time in the fitted curve; and (iv) the error term (e_i_) [24]. A least-squares cosine wave was fitted to the data to test for the presence and characteristics of circadian rhythms. The circadian rhythm was validated and quantified using the single cosinor method, which is a procedure of periodic regression analysis for fitting a 24 h cosine curve to raw data. Cosinor analysis was conducted based on the LME model framework to estimate the MESOR, amplitude, and acrophase of the circadian model. Statistical significance is determined for each of the given metrics by an F test with respect to the null hypothesis (zero amplitude or no rhythm). Cosinor analysis was adjusted for age and sex due to the relationship between tongue features and these variables [25]. Statistical analyses were performed using R statistical software version 4.1.1 (30 September 2021). The statistical significance in the analysis was set to 0.05.

## 3. Results

### 3.1. Demographic Characteristics

Finally, fifteen healthy laboratory subjects participated in this clinical study to analyze their tongue features and circadian rhythms. Data for one participant were excluded from the dataset because of voluntary withdrawal.

The baseline characteristics of the fourteen participants are presented in Table 2. The participants ranged from 20 to 69 years. All fourteen participants had normal blood pressure levels (systolic blood pressure < 140 mmHg, diastolic blood pressure > 60 mmHg).

The mean age of the fourteen participants was 39.7 ± 20.7 years. No differences were found in age, weight, or BMI between the males and females. In addition, no significant differences were observed in diastolic blood pressure, pulse rate, or temperature between the two groups. However, height and systolic blood pressure were significantly higher in the female group than in the male group (*p* = 0.000 and 0.045).

### 3.2. Adverse Events

After the acquisition of tongue images, we assessed safety issues. No adverse events were reported in this study. In addition, no adverse events not related to the investigational device were observed during or after the clinical study.

### 3.3. Circadian Variation in Tongue Features

Table 3 shows the mean and standard deviation of the MESOR, 95% confidence interval (95% CI) of amplitude, and acrophase of the CIE Lab color value in the tongue body area. There was a significant difference in the circadian rhythm for all variables (*p* < 0.001). In particular, the amplitudes of the CIE a* and b* color values were significantly different (*p* = 0.0472 and 0.0475). The circadian crests of all the variables were synchronous. Details on the circadian variation in the CIE Lab color values of the tongue body are presented in Table 3 and are shown in Figure 3.

The minimum CIE L* in the tongue body area occurred from 14:00 to 16:00. In contrast, the maximum CIE L* in the tongue body area was estimated to occur from 01:00 to 04:00. Similar to the CIE L* value, the minimum CIE a* in the tongue body area occurred from 13:00 to 16:00, and the maximum was estimated to occur from 01:00 to 04:00. The minimum of the CIE b* in the tongue body area occurred from 04:00 to 07:00. In contrast, the maximum of the CIE b* in the tongue body area was estimated to occur from 16:00 to 19:00.

Details on the circadian variation in the tongue coating color values are presented in Table 4 and are shown in Figure 4. Table 4 shows the mean and standard deviation of the MESOR, 95% confidence interval (95% CI) of amplitude, and acrophase of the CIE Lab in the tongue coating area. The tongue coating color values showed significant circadian rhythms (*p* < 0.001). In particular, the amplitudes of the CIE b* color values were significantly different (*p* < 0.005). The circadian crests of all the variables were synchronous.

The minimum CIE L* in the tongue coating area occurred at approximately 16:00. In contrast, the maximum CIE L* in the tongue coating area was approximately 04:00. The minimum CIE a* in the tongue coating area occurred from 12:00 to 14:00. In contrast, the maximum CIE a* in the tongue coating area was estimated to occur from 00:00 to 02:00. The minimum CIE b* in the tongue coating area occurred from 07:00 to 10:00. In contrast, the maximum of the CIE b* in the tongue coating area was estimated to occur from 19:00 to 22:00.

Details on the circadian variation in the tongue coating percentage are presented in Table 5, which shows the mean and standard deviation of the MESOR and the 95% confidence intervals (95% CIs) of the amplitude and acrophase of the tongue coating percentage. Overall, the tongue coating percentage showed significant circadian variations (*p* < 0.001). The circadian crests of all the variables were synchronous.

The tongue coating percentage showed significant circadian variations, as shown in Figure 5. The minimum tongue coating percentage occurred from 14:00 to 16:00, whereas the maximum tongue coating percentage in the whole area occurred from 01:00 to 04:00. The peak amplitude occurred at approximately 04:00 (*p* = 0.0311).

## 4. Discussion

The results of the present study showed that circadian changes exist in tongue features. In particular, the tongue coating percentage showed significant differences in both amplitude and acrophase.

Previous investigations have shown that the salivary flow rate fluctuates with circadian rhythm. It has been suggested that the unstimulated salivary flow rate may be at its maximum in the midafternoon. C. Dawes reported that the unstimulated salivary flow rate showed very marked circadian rhythms of high amplitude with an acrophase at 15:26 h [8]. The results of previous studies are consistent with the results of this study in that the tongue coating percentages in the whole area showed extremely low amplitudes from 14:00 to 16:00, whereas they showed the highest amplitudes from 01:00 to 04:00. The CIE L* color value of the tongue coating area, which is highly correlated with the tongue coating percentage, also showed consistent results.

The results of this study showed that the tongue coating percentage significantly differed before and after sleep and meals (Table 6). The percentage of coatings on the tongue was significantly lower after meals than before meals. After sleep (the following morning at 7:00 a.m.), the tongue coating percentage was significantly greater than that before sleep, indicating that the effects of sleep interacted with those of the circadian pacemaker. Sleep markedly enhanced the increase in tongue coating percentage but did not influence tongue coloration. A decrease in oral activity and a decrease in saliva production, together with an increase in the proliferation of bacteria during sleep, may increase the use of a tongue coating. In particular, increased tongue coating may be closely related to several factors, including reduced saliva production and decreased oral activity during the night, as part of a well-known circadian cycle. Comparative analysis before and after a meal and during sleep was performed for all tongue features. A comparison of the tongue features before and after a meal and sleep revealed that the percentage of patients with a tongue coating changed significantly less after a meal and significantly more than those after sleep.

The tongue coating is dependent on microcirculation, which changes with body temperature, hormone levels, fluid circulation, and hydrodynamic parameters. Although the change in tongue coating amount may be caused by the interaction of multiple factors [13], it is possible that the increase or decrease in the salivary flow rate could be the main factor contributing to the increase or decrease in tongue coating amount or thickness, respectively. A review of previous studies revealed that the presence of a tongue coating was strongly related to the resting saliva flow rate. The saliva is divided into non-stimulated whole saliva, stimulated saliva (whole saliva, ductal secretion, parotid glands, submandibular glands, and sublingual glands), and direct secretion (parotid gland, submandibular glands, and sublingual glands). The stimulated saliva is secreted more by food or oral stimulation, and non-stimulated whole saliva is secreted more during sleep or when there is no oral irritation. It is well known that during sleep, the salivary flow rate is extremely low. According to previous studies, the salivary flow rate decreases markedly during sleep; on the other hand, the salivary flow rate increases markedly during intake [11,12]. Shinjiro Koshimune, DDS. reported that this study showed an extreme reduction in the resting saliva flow rate when the amount of tongue coating increased [13]. On the other hand, participants with an extremely low resting saliva flow rate had a significantly greater tongue coating score than the other participants. Therefore, the results of this study suggested that a reduction in salivary flow might influence the production of tongue coating. Several previous studies reported opposite results to those of this study. In contrast, Radojkova-Nikolovska Vera. et al. reported that there was no correlation between unstimulated salivary flow and tongue coating [26].

In addition to the factors mentioned above, oral activity may have had an effect on tongue body color and tongue coating percentage. Oral activity during a meal maximizes tongue movement and increases blood flow, which can make the tongue redder and wash out the tongue coating on the tongue surface.

The circadian rhythm was validated and quantified using the single cosinor method, which is a procedure of periodic regression analysis for fitting a 24 h cosine curve to raw data. Equation (1) depicts the regression model for the tongue coating percentage in the whole area. The single cosinor model of tongue coating percentage in the whole area can be described by the following formula:(1)$$x_{t}=20.378+6.263\cos\left(2\pi t/24+0.879\right)+e_{i}$$
where

*x_t_* represents the mean data collected at times *t*;

*e_i_* denotes the error term at each time, assumed to be independent, normally distributed, with a mean of zero and unknown constant variance σ^2^.

Although the results of this study were based on weak evidence, they might be useful for providing essential information for clinical studies related to tongue diagnosis or oral conditions. In addition, it is possible to make a more accurate tongue diagnosis through correction based on the regression model of tongue features, and if the regression model is developed with a large amount of data, it is highly likely to be used as reference data for the diagnosis. The results also suggest the necessity of conducting a study on circadian variations in tongue features, specifically targeting diseases such as insomnia, heart disease, diabetes, Sjögren’s syndrome, and xerostomia, all of which are closely associated with tongue features.

The three limitations of the current study were as follows. First, this study was limited by the small number of participants and its single-center design. The results are very difficult to generalize and standardize, and the clinical usefulness and applicability of the results of this study remain questionable. However, various studies, including case–control studies and randomized controlled clinical trials in multiple centers, are needed. Second, additional studies are required to assess each correlation and the changes between clinical factors (such as salivary flow rate, oral temperature, oral components, and hormones) and tongue features according to the circadian rhythm. Third, even though the experimental setup for measurements over 24 h has been shown to be feasible, it is still in its early stages, and further improvements are needed. However, despite the above limitations, this is very meaningful because it is the first study to verify circadian variations in tongue features.

## 5. Conclusions

This study revealed changes in tongue features according to circadian rhythm for the first time. The colors of the tongue body, tongue coating, and percentage of tongue coating showed significant circadian variations. The tongue is a unique organ, and the oral system is directly exposed to the external environment. In addition, the saliva, microorganisms, and food particles remain on the tongue surface long-term. As is known, multiple factors such as dietary habits, sleep, and micro-ecological environment [27] are involved in the formation of tongue coating and change of tongue color. Similarly, these factors are reflected in oral health [28]. Above all, these aspects support that this circadian rhythm has important clinical implications for the timing of tongue diagnosis. The majority of evidence from this study should consider the time of day at which the tongue is observed and the evaluation of tongue status.

## Figures and Tables

**Figure 1 jcm-13-03549-f001:**
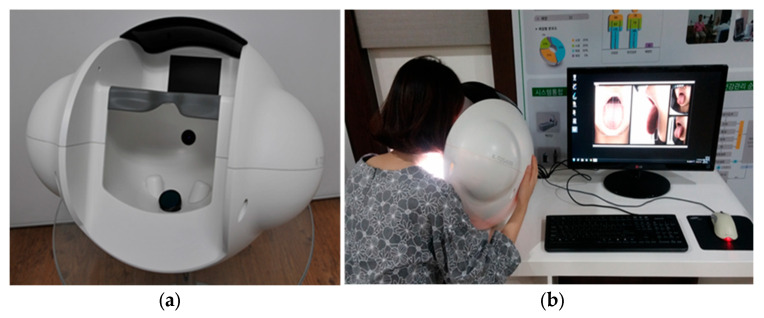
The investigational computerized tongue image acquisition system (**a**) and a scene in which a tongue image is acquired with the investigational device (**b**).

**Figure 2 jcm-13-03549-f002:**
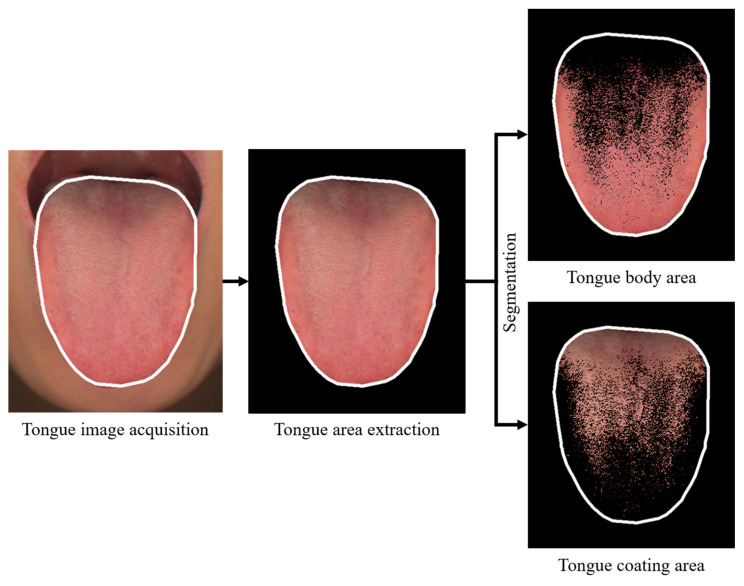
Process of tongue image extraction. The tongue area was extracted from an acquired tongue image. The tongue coating area was distinguished from the tongue body area based on differences in color. The tongue coating percentage was calculated as the percentage of the pixel number of the tongue coating area to the pixel number of the whole tongue area.

**Figure 3 jcm-13-03549-f003:**
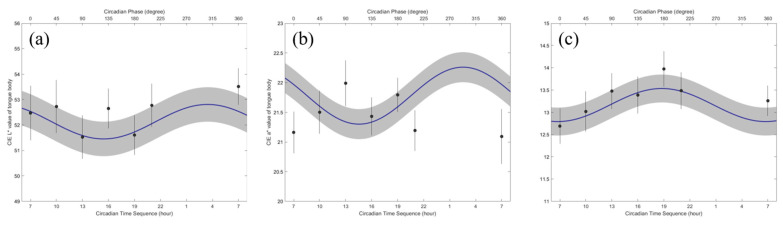
The cosinorgram of the CIE L* color value (**a**), CIE a* color value (**b**), and CIE b* color value (**c**) in the tongue coating area. The circadian rhythm and standard error bands were estimated from the single cosinor procedure. The circles and lines indicate the least squared mean and standard error at each circadian hour.

**Figure 4 jcm-13-03549-f004:**
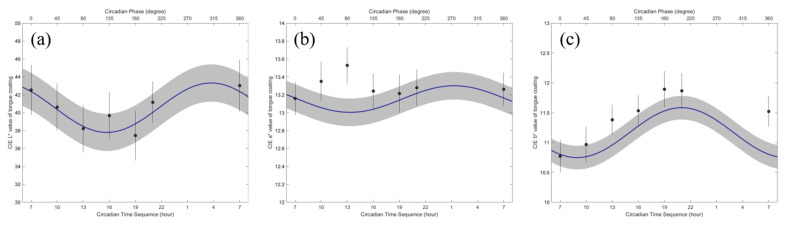
The cosinorgram of the CIE L* color value (**a**), CIE a* color value (**b**), and CIE b* color value (**c**) in the tongue coating area. The circadian rhythm and standard error bands were estimated from the single cosinor procedure. The circles and lines indicate the least squared mean and standard error at each circadian hour.

**Figure 5 jcm-13-03549-f005:**
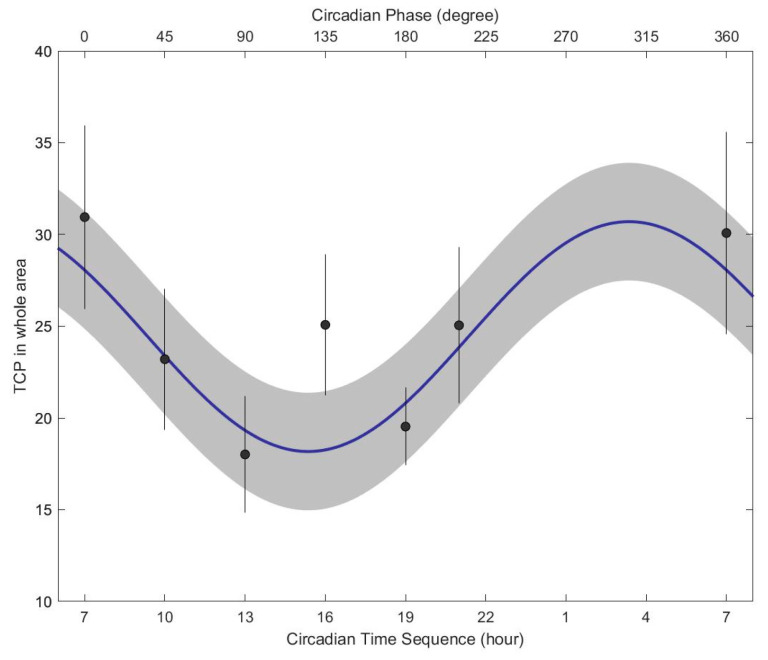
Cosinorgram of the tongue coating percentage in the whole tongue area. The circadian rhythm and standard error bands were estimated from the single cosinor procedure. The circles and lines indicate the least squared mean and standard error, respectively, at each circadian hour.

**Table 1 jcm-13-03549-t001:** Clinical experimental procedure.

Visit	Visit 1	Visit 2	Visit 3						Visit 4
Day	−4 W–0D	1D	2D	2D	2D	2D	2D	2D	3D
Epoch	Screening	Stabilization	Measurement						
Time	09:00–12:00	16:00–24:00	07:00	10:00	13:00	16:00	19:00	22:00	07:00
Acquire consent from the subject	✓								
Selection and exception criteria	✓								
Demographical information	✓								
Medical history	✓								
Laboratory test (screening)	✓								
Subject training	✓	✓	✓	✓	✓	✓	✓	✓	✓
Laboratory test (evaluation)			✓					✓	✓
Vital signs	✓	✓	✓	✓	✓	✓	✓	✓	✓
Acquisition of a tongue image		✓	✓	✓	✓	✓	✓	✓	✓
Adverse event		✓	✓	✓	✓	✓	✓	✓	✓

Meal time: Visit 2 (18:00–18:30), Visit 3 (8:00–8:30, 12:00–12:30, 18:00–18:30), Visit 4 (N/A); sleep time: 23:00–07:00 (8 h).

**Table 2 jcm-13-03549-t002:** Chronobiometric parameters for the CIE Lab color values in the tongue body.

Variables	Male (n = 9)	Female (n = 5)	*p* Value
Age (years)	41.116 ± 21.684	37.082 ± 20.809	0.741
Height (cm)	169.656 ± 4.494	158.740 ± 3.018	0.000 **
Weight (kg)	69.467 ± 13.655	56.160 ± 6.750	0.066
Body mass index (kg/cm^2^)	24.033 ± 4.052	22.343 ± 3.215	0.440
Systolic blood pressure (mmHg)	129.667 ± 9.314	116.200 ± 13.312	0.045 *
Diastolic blood pressure (mmHg)	78.333 ± 12.227	70.000 ± 5.339	0.104
Pulse rate (bpm)	72.556 ± 9.645	76.200 ± 7.887	0.486
Temperature (°C)	36.811 ± 0.276	36.640 ± 0.219	0.258

Data were presented as the mean ± standard deviation and compared by independent *t* test. The statistical significance was indicated by following notations: * *p* value < 0.05 and ** *p* value < 0.01.

**Table 3 jcm-13-03549-t003:** Chronobiometric parameters for the CIE Lab color values in the tongue body.

Variable	MESOR	Amplitude (95% CI)	Acrophase (95% CI)
CIE L*	50.796 (0.992)	0.678 (−0.518–1.874)	0.900 (−0.463–2.264)
CIE a*	23.028 (0.370)	0.480 (0.006–0.954)	0.665 (0.027–1.303)
CIE b*	12.702 (0.459)	0.372 (0.004–0.739)	−1.373 (−2.973–0.228)

**Table 4 jcm-13-03549-t004:** Chronobiometric parameters for the CIE Lab color values for the tongue coating.

Variable	MESOR	Amplitude (95% CI)	Acrophase (95% CI)
CIE L*	39.540 (3.046)	2.757 (−0.822–6.337)	0.974 (−0.098–2.046)
CIE a*	12.878 (0.225)	0.148 (−0.147–0.444)	0.343 (−0.836–0.569)
CIE b*	10.280 (0.290)	0.418 (0.165–0.671)	−0.793 (−1.660–0.074)

**Table 5 jcm-13-03549-t005:** Chronobiometric parameter for the tongue coating percentage (%).

Variable	MESOR	Amplitude (95% CI)	Acrophase (95% CI)
Tongue coating percentage	20.378 (4.691)	6.263 (0.569–11.958)	0.879 (0.189–1.569)

**Table 6 jcm-13-03549-t006:** Comparison of tongue coating percentage before and after meals and sleep (N = 14).

	Before	After	Differences (ΔE)	*p*-Value
Morning meal	30.95 (18.68)	23.20 (14.41)	−7.75 (10.26)	0.025 ^‡^*
Afternoon meal	23.20 (14.41)	18.02 (11.88)	−5.19 (8.62)	0.049 ^‡^*
Evening meal	25.08 (14.38)	19.53 (7.93)	−5.55 (8.14)	0.024 ^†^*
Sleep	25.05 (15.89)	30.08 (20.68)	5.03 (7.19)	0.021 ^†^*

Data were presented as mean ± standard deviation. Data were compared using a paired *t*-test or Wilcoxon signed-rank test. The statistical significance was indicated by following notations: * *p*-value < 0.05. ^†^
*p*-values were derived from the paired *t*-test. ^‡^
*p*-values were derived from the Wilcoxon signed-rank test.

## Data Availability

The datasets used and/or analyzed during the current study are available from the first author upon reasonable request.

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
