# Peer review of "Circadian Rhythms in Tongue Features"

_jcm, 2024, doi:10.3390/jcm13123549_

Round 1

Reviewer 1 Report

Comments and Suggestions for Authors

General Comments:

The manuscript by Jihye Kim et al. titled "Circadian Rhythms in Tongue Features" presents an interesting study determining the circadian variations in tongue features such as body color, coating color, and coating thickness in healthy participants. The use of cosinor analysis to quantify these rhythms is commendable and contributes novel insights into the temporal dynamics of oral health. However, there are several areas where the manuscript could be strengthened to improve the robustness of the findings.

Major Concerns:

1.      The research had a limited number of only fifteen participants, which is likely to not give enough power to discover minor differences in features of the tongue or even to extend the findings to a wide population. Widening the sample size and acquiring participants with different backgrounds will strengthen the reliability and practicality of the outcomes.

2.      The manuscript would benefit from more detailed exposition on the image acquisition protocol and analysis techniques. Specifically, readers are left wanting for details regarding calibration of the computerized tongue image acquisition system— how was it done? What were the settings used during image capture?

 3.      The information provided is unclear about whether such factors as recent food intake; oral hygiene practices or medications influencing oral physiology were properly controlled or documented. More rigid control of these variables will ensure that observed variations are mainly due to circadian rhythms. Moreover, the participants' age group varies significantly from 20 to 69 years old. It will be worthwhile to determine whether age have an effect on circadian rhythms in tongue features.

 4.      Cosinor analysis is suitable, but the manuscript should include a discussion of any model fit statistics or other models considered in validation of choice of analysis method. Moreover, given the number of endpoints analyzed adjustments for multiple comparisons would strengthen the statistical conclusions.

Minor Concerns:

1.      Literature Review: Although the introduction provides a good background on circadian rhythms, it could be expanded to include more specific studies related to oral circadian rhythms— especially those involving tongue features—to better justify the objective of the current study within the existing research landscape.

2.      The clinical implications of this study should be extensively discussed such as how could these findings influence the scheduling of clinical assessments or oral health diseases' control?

3.      The manuscript could benefit from proofreading to correct minor grammatical errors and enhance clarity. This includes the consistent use of terminology and clearer descriptions of the study procedures.

 The reference list should be updated to include some new relevant citations from the years 2023 and 2024. Currently very old references of 1965 and 1972 are cited in the manuscript which are very outdated.

Comments on the Quality of English Language

The manuscript could benefit from proofreading to correct minor grammatical errors and enhance clarity. This includes the consistent use of terminology and clearer descriptions of the study procedures.

Author Response

Comments 1: The research had a limited number of only fifteen participants, which is likely to not give enough power to discover minor differences in features of the tongue or even to extend the findings to a wide population. Widening the sample size and acquiring participants with different backgrounds will strengthen the reliability and practicality of the outcomes.

Response 1: I completely concur with this comment. While it indeed represents a limitation of our study, it's important to recognize that our research represents a pioneering effort in validating circadian variations in tongue features. These aspects, along with the noted limitations, have been duly addressed in our manuscript. (section 4. Discussion p. 12, lines 400-11).

Comments 2: The manuscript would benefit from more detailed exposition on the image acquisition protocol and analysis techniques. Specifically, readers are left wanting for details regarding calibration of the computerized tongue image acquisition system— how was it done? What were the settings used during image capture?

Response 2: Thank you for your kindly comments.

1) I have added the detailed the extraction and calculation methods of tongue feature in manuscript as follows (section 2.5. Extraction and calculation of tongue features, p.5, lines 187-201):

A whole tongue area was automatically segmented using the combined polar edge method and the gradient vector flow snake technique [21]. For incorrect segmentations, both manual and automatic segmentation were performed repetitively. A dark region in the tongue root area, because of low illumination intensity, and high-luminance regions, caused by light reflection from saliva, were removed using thresholding with CIE L* color values, as these colors could introduce errors into the color histogram analysis. The region of the tongue was extracted from the captured image, distinguishing the tongue coating area from the tongue body area based on the variance between their respective RGB color values. Subsequently, obtained RGB color values of each pixel in both areas were convert-ed to CIE Lab color values. The mean CIE Lab color values of the tongue body and the tongue coating area were then computed. The tongue coating percentage was determined by calculating the ratio of the pixel count of the tongue coating area to the total pixel count of the entire tongue area. The process for calculating these tongue indices is illustrated in Figure 2. The tongue image extraction methods and image analysis methods for tongue features in CTIS have been described in detail in previous studies [21,22].

2) The details regarding calibration of the CTIS were described in manuscript (section 2.4. Measurement, p.4, lines 151-7).

: The CTIS consist of tongue positioning part, lighting part, image acquisition part, and data processing part (example. analysis software). To minimize color errors, the image ac-quisition part was calibrated using a color checker (X-Rite, Grand Rapids, MI, USA) before capturing the tongue images. The accuracy of color calibration was assessed by calculat-ing measurement errors between gold standards (true color values) and measurement color values of color checker. A low measurement errors indicates the high accuracy of the color measurement [19,20].

3) The settings used during image capture were as follows: (section 2.4. Measurement, p.4, lines 158-62).

The image acquisition part can capture the tongue image at a resolutions of 1080 x 1440 pixels and automatically adjust white balance, exposure time, and gain settings. The lighting part should meet the following conditions: an illuminance of 500 lux, a color temperature over 5000 Kelvin, a color rendering index greater than 90, and a min/max ratio of illuminance distribution under 0.9.

Comments 3: The information provided is unclear about whether such factors as recent food intake; oral hygiene practices or medications influencing oral physiology were properly controlled or documented. More rigid control of these variables will ensure that observed variations are mainly due to circadian rhythms. Moreover, the participants' age group varies significantly from 20 to 69 years old. It will be worthwhile to determine whether age have an effect on circadian rhythms in tongue features.

Response 3: Agree.

1) I excluded participants who were taking medications or had diseases that influence oral physiology based on the exclusion criteria. Additionally, participants were instructed to adhere to the following criteria: (1) refraining from using any oral rinse for 1 week preceding the experiment day; (2) avoiding consumption of caffeine, alcohol, milk, and smoking for 24 hours prior to the experiment; and (3) abstaining from eating and drinking for a minimum of 8 hours before the experiment, although they were permitted to drink water up to 3 hours before the experiments. These measures were implemented to minimize the influence of various factors (pp. 4-5, lines 165-71 and p.3, lines 114-5).

2) Agree. Therefore, I performed that cosinor analysis was adjusted for age and sex due to the relationship between tongue features and these variables. That point was described in manuscript (section 2.6. Statistical analysis, p. 7, lines 242-3).

Comments 4: Cosinor analysis is suitable, but the manuscript should include a discussion of any model fit statistics or other models considered in validation of choice of analysis method. Moreover, given the number of endpoints analyzed adjustments for multiple comparisons would strengthen the statistical conclusions.

Response 4: I agree. Considering multiple comparisons during statistical analysis is crucial. Cosinor analysis comprehensively addresses all the points you highlighted.

Comments 5: Literature Review: Although the introduction provides a good background on circadian rhythms, it could be expanded to include more specific studies related to oral circadian rhythms— especially those involving tongue features—to better justify the objective of the current study within the existing research landscape.

Response 5: Unfortunately, no reports have been published regarding changes in tongue features, such as tongue color or tongue coating thickness, according to circadian rhythm. Therefore, I have added references regarding the association between tongue features and the menstrual cycle (section 1. Introduction, p. 2, lines 72-80).

: Although this is not a study on circadian changes, there is research on variations in tongue features, such as tongue color and tongue coating thickness, according to the menstrual cycle [16,17]. Kim, J., et al. revealed that the CIE Lab color value in the tongue coating area and the tongue coating thickness of primary dysmenorrhea patients during the menstrual phase were significantly lower than those of the non-dysmenorrhea patients [16]. Hsieh, S.-F., et al. reported that tongue color changes during the menstrual cycle and suggested that there are differences in tongue color between the follicular and luteal phases among childbearing women [17].

Comments 6: The clinical implications of this study should be extensively discussed such as how could these findings influence the scheduling of clinical assessments or oral health diseases' control?

Response 6: Thank you for your opinions. I revised the conclusions based on the comments (p. 12, lines 415-21)

: The tongue is a unique organ, and the oral system is directly exposed to the external environment. In addition, the saliva, microorganism and food particles remain on the tongue surface long-term. As is known, multiple factors such as dietary habits, sleep, and micro-ecological environment [27] are involved in the formation of tongue coating and change of tongue color. Similarly, these factors are reflected on the oral health [28]. Above all, these aspects supported that this circadian rhythm has important clinical implications for the timing of tongue diagnosis.

Comments 7: The manuscript could benefit from proofreading to correct minor grammatical errors and enhance clarity. This includes the consistent use of terminology and clearer descriptions of the study procedures.

Response 7: The paper has been proofread in English using AJE service.

Comments 8: The reference list should be updated to include some new relevant citations from the years 2023 and 2024. Currently very old references of 1965 and 1972 are cited in the manuscript which are very outdated.

Response 8: I wholeheartedly acknowledge your point. Regrettably, due to the absence of recent comparative studies, we were compelled to rely on older references for our research findings.

Reviewer 2 Report

Comments and Suggestions for Authors

Jihye Kim and co-worker presented an insightful study entitle "Circadian Rhythms in Tongue Features" where the study aimed to investigate the circadian rhythms of tongue features according to the effects of physiological phases over a 24 h period, where Cosinor analysis revealed that all tongue features were significantly related to circadian rhythm. Moreover, the limitation indicated considering the overall results made the draft more insightful; however still some suggestions listed below needs to be addressed for making the paper for wider readers. 

1. Suggested to add a graphical abstract that demonstrate the title of the investigation.

2. Suggested to explain why the research has been kept for more than six year unpublished and what happen/made to publish it after so long gap, since the work was September 2015 to September 2016.

3. Suggested to correct the grammar "The consumption of highly colored foods and sweets will affect tongue color."

4. Suggested to look whether calculating total color difference and ΔE using CIE L, a, and b values if some more useful information can be obtained to make circadian rhythm discussion briefly.

Author Response

Comments 1: Suggested to explain why the research has been kept for more than six year unpublished and what happen/made to publish it after so long gap, since the work was September 2015 to September 2016.

Response 1: The research period experienced interruptions due to department transfers and personal reasons, including IVF, pregnancy, childbirth, and a 5-year leave of absence for childcare. This manuscript marks my first significant achievement since returning.

Comments 2: Suggested to correct the grammar "The consumption of highly colored foods and sweets will affect tongue color.

Response 2: I changed the sentence as follows: The consumption of highly colored foods and sweets can affect the tongue color.

 (section 2.2. Participants and enrollment, p. 4, line 137)

Comments 3: Suggested to look whether calculating total color difference and ΔE using CIE L, a, and b values if some more useful information can be obtained to make circadian rhythm discussion briefly.

Response 3: I have added a table illustrating the differences (ΔE) in tongue coating percentage before and after meals and sleep. Notably, the tongue coating percentage exhibited significant differences in both amplitude and acrophase, indicating a remarkable circadian rhythm.

(section 4. Discussion, p.11, line 351)

Table 6. Comparison of tongue coating percentage before and after in meals and sleep (N = 14).

Before

After

Differences (ΔE)

P-value

Morning meal

30.95 (18.68)

23.20 (14.41)

-7.75 (10.26)

0.025*

Afternoon meal

23.20 (14.41)

18.02 (11.88)

-5.19 (8.62)

0.049*

Evening meal

25.08 (14.38)

19.53 (7.93)

-5.55 (8.14)

0.024*

Sleep

25.05 (15.89)

30.08 (20.68)

5.03 (7.19)

0.021*

Data were presented as mean ± standard deviation. Data were compared using a paired t-test or Wilcoxon signed-rank test. The statistical significance was indicated by following notations: * P-value < 0.05 and ** P-value < 0.01.

† P-values were derived the paired t-test.

‡ P-values were derived the Wilcoxon signed-rank test.

Round 2

Reviewer 2 Report

Comments and Suggestions for Authors

The authors have reflected all the said suggestions and comments, which made the manuscript enhanced with improved readability; Thus I suggest for further consideration with acceptance.